# Struggles and strategies in anaerobic and aerobic cycling tests: A mixed-method approach with a focus on tailored self-regulation strategies

**Anna Hirsch**[1]*, **Maik Bieleke**[1], **Raphael Bertschinger**[1], **Julia Schüler**[1], **Wanja Wolff**[1,2]

**1** Department of Sport Science, Sport Psychology, University of Konstanz, Konstanz, Germany,
**2** Educational Psychology, Institute of Educational Science, University of Bern, Bern, Switzerland

\* anna.hirsch@uni-konstanz.de

**Data Availability Statement:** All data files are available at osf.io/mq6kt/.

**Funding:** Zukunftskolleg at the University of Konstanz, https://www.uni-konstanz.de/

## Abstract

Endurance sports pose a plethora of mental demands that exercisers have to deal with. Unfortunately, investigations of exercise-specific demands and strategies to deal with them are insufficiently researched, leading to a gap in knowledge about athletic requirements and strategies used to deal with them. Here, we investigated which obstacles exercisers experience during an anaerobic (Wingate test) and an aerobic cycling test (incremental exercise test), as well as the strategies they considered helpful for dealing with these obstacles (qualitative analysis). In addition, we examined whether thinking of these obstacles and strategies in terms of if-then plans (or implementation intentions; i.e., "If I encounter obstacle O, then I will apply strategy S!") improves performance over merely setting performance goals (i.e., goal intentions; quantitative analysis). $N = 59$ participants (age: $M = 23.9 \pm 6.5$ years) performed both tests twice in a 2-within (Experimental session: 1 vs. 2) × 2-between (Condition: goal vs. implementation intention) design. Exercisers' obstacles and strategies were assessed using structured interviews in Session 1 and subjected to thematic analysis. In both tests, feelings of exertion were the most frequently stated obstacle. Motivation to do well, self-encouragement, and focus on the body and on cycling were frequently stated strategies in both tests. There were also test-specific obstacles, such as boredom reported in the aerobic test. For session 2, the obstacles and strategies elicited in Session 1 were used to specify if-then plans. Bayesian mixed-factor ANOVA suggests, however, that if-then plans did not help exercisers to improve their performance. These findings shed novel light into the mental processes accompanying endurance exercise and the limits they pose on performance.

## Introduction

The specific psychological obstacles that exercisers face during an endurance performance and the strategies they apply to deal with them requires more research as this topic is of great scientific and practical importance [1, 2], for example when it comes to enhancing motivation on continuous athletic activity or exercisers' athletic performance in general. To understand the

zukunftskolleg/ University of Konstanz open access publication funds German Federal Institute for Sports Science (BISp; Aktenzeichen: 071003/19-20), https://www.bisp-sportpsychologie.de/SpoPsy/DE/Home/home_node.html The funders had no role in study design, data collection and analysis, decision to publish, or preparation of the manuscript.

**Competing interests:** The authors have declared that no competing interests exist.

limits of endurance performance (i.e., factors that hamper or disrupt endurance performance) and design effective sport psychological interventions, knowledge about the various exercise-related psychological obstacles that might affect performance is crucial [3]. Accordingly, a growing body of research focuses on the obstacles that endurance athletes have to deal with during performance (e.g., [1, 4]) and comprehensive reviews shed light on the characteristics of psychological interventions in sport that allow them to improve endurance performance (e.g., [5]).

According to the psychobiological model of exercise tolerance [6, 7], the perceived exertion (referred to as "the conscious sensation of how hard, heavy, and strenuous a physical task is"; [8]) plays a critical role: When the application of additional effort appears unreasonable or not possible, endurance performance (i.e., cycling) is terminated [6, 7]. Other frameworks like the taxonomy of fatigue argue that perceived fatigability (referred to as "the sensations that regulate the integrity of the performer based on the maintenance of homeostasis and the psychological state of the individual"; [9]) is impacted by exercise-induced sensations like exertion or pain and thus influences performance fatigability (i.e., muscle and nervous system capacity to generate a sufficient response for a given exercise; [9]). Exercise-induced pain itself is also an important determinant of endurance performance, as higher pain tolerance is associated with enhanced endurance performance (see [10] for a detailed overview). This is consistent with observations that former Olympic cyclists report successful coping with pain to be the greatest psychological challenge [11], and that being occupied with physical discomfort impaired performance [4]. Corroborating the assumptions made in these theoretical frameworks, a focus group study with recreational endurance exercisers revealed that exercise-related sensations like exertion, pain, fatigue, or discomfort were the most common challenges [1]. Beyond dealing with these bodily sensations, staying focused and motivated after difficult situations are further obstacles recreational exercisers report. However, sport-specific obstacles differ between athletes and non-athletes: For example, non-athletes are more likely to process physiological signals during sports negatively (e.g., sensations of exertion and pain) which might in turn trigger negative affective reactions [12]. Also, non-athletes reported more pain, discomfort, or articulated irrelevant information for the exercise than athletes [13]. Thus, lack of experience creates critical obstacles in sport.

Beyond focusing on perceived obstacles, it is also crucial to know which strategies exercisers spontaneously apply to deal with them. Deliberately implemented strategies and programs have been studied intensively (e.g., [2, 14]) and show that endurance performance can be enhanced by mental imagery, self-talk, and goal setting [2]. For instance, self-talk interventions can reduce perceptions of effort as one determinant of endurance performance [15]. Active distraction at low training intensities was perceived as helpful in a sample of recreational runners to reduce boredom and increase positive emotions [16]. Accordingly, metacognitive skills -pertaining to "planning and reviewing one's attentional focus" [4], seem to be helpful for athletes in order to gain knowledge about the use of cognitive strategies. As with exercise-related obstacles, expertise and fitness level affect the kind of strategies exercisers use [17]. Thus, a key characteristic of a strategy to enhance endurance performance is the adaptability to requirements and demand levels of different sports and athletes' abilities. To our knowledge, research that elicits the specific obstacles exercisers face during a single bout of exercise, and that assesses the strategies exercisers use to cope with these challenges is lacking.

## Obstacles and strategies might depend on exercise characteristics

Exercise-induced obstacles and strategies to deal with them vary not only as a function of the exercisers' training level and experience but also as a function of the task demands associated

with an exercise. Within a given sport, physiological demands vary in terms of changes in neural blood circulations and metabolism [18], reduced oxygen distribution [19], and ionic and metabolic modifications in the concerned muscle [20]. Consequently, these effects also affect our psychological response to athletic activities. For instance, our affective response is modulated by the intensity of an exercise [21]: At moderate intensities (i.e., up to the aerobic threshold) exerciser tend to have positive affective response, whereas at very high intensities (i.e., above the anaerobic threshold) the affective response tends to be negative. This difference in affective response has been explained (among others) by the different interoceptive sensations created by these intensities [21]. The three-dimensional framework of perceived fatigability further highlights the differential contributions of sensory-discriminatory (i.e., perception of strain), affective-motivational (i.e., key emotion), and cognitive-evaluative components (i.e., mindset) to endurance performance [22]. For example, a short anaerobic test can be completed by most exercisers, as it covers an exercising time of only 30 seconds, which might lead to a higher sense of achievability. However, the short duration of the test means that there is little time to adjust performance strategies. An aerobic test, on the other hand, is open ended and will therefore lead to complete exhaustion of the exerciser, likely leading to negative affective reactions. It is each participant's decision when to terminate the test, which also makes performance dependent on motivation and attitude. Due to the steady increase in power output, the aerobic test probably requires significantly more self-control from the exerciser (see [23] for a similar reasoning). Due to its longer duration, however, participants might also have more opportunities to adapt their strategies in order to last longer. Thus, different exercises (within one type of sport, as well as across sports) vary in the physiological and psychological demands they impose, which in turn might create obstacles that are very exercise-specific. To account for these differences, strategies need to be tailored to the obstacles an exerciser faces [24].

## Optimizing performance with tailored self-regulatory interventions

Optimal performance hinges on effective self-regulation (i.e., "the capacity of organisms (here, human beings) to override and alter their responses" [25]), the importance of which for sports performance is well established (e.g., [26]). Beyond relying on their intuitive lay knowledge, exercisers might benefit from employing targeted self-regulatory strategies to deal more effectively with their exercise-induced obstacles. One promising self-regulatory strategy is if-then planning (often referred to as *implementation intentions;* [27]). Implementation intentions are an effective self-regulation strategy in a variety of contexts [28, 29], like in the context of physical exercise [30], health (e.g., [31–33]), and researchers have started to investigate its effects on endurance performance (e.g., [34]). If-then planning is based on establishing links between goal-relevant situations (e.g., obstacle: experiencing the urge to stop) and goal-directed behaviors (e.g., overcoming the obstacle: cheering yourself on) in an if-then format: "If I encounter obstacle X, then I will perform behavior Y!" (e.g., "And if I feel the urge to stop, then I tell myself: You can do it!"; [35]). The structure of if-then plans is assumed to facilitate goal attainment by strengthening the mental representation of the goal-relevant situation, making it more accessible and easier to recognize [27, 36]. Moreover, if-then planning is assumed to automate the initiation of the goal-directed behavior [37].

If-then planning is typically contrasted with forming goal intentions, which refers to merely specifying a desired outcome or a desired behavior (e.g., "I want to keep going as long as possible!"; [38]). Because such goals comprise no specific link between a situation and a behavior, they are conducive to more deliberative, top-down ways of self-regulation than if-then planning, which facilitates automatic, bottom-up ways of self-regulation. Importantly, research

shows that when it comes to staying on track, even when performing the goal-directed behavior is perceived as aversive, if-then planning is a more effective self-regulation strategy than goal setting [39]. Moreover, if-then plans are effective for counteracting impulsive reactions [40], which can be helpful for example when an athlete has to control the impulse to follow every acceleration of her opponents (because not controlling this impulse would eventually wear out the athlete, reducing the chance to win the race). Their desirable cognitive mechanisms (e.g., automaticity) and their effectiveness across various domains (e.g., health, exercise) render if-then plans a promising self-regulatory strategy in endurance sports [41]. However, prior research in this domain has so far produced mixed results (see [34] for a comprehensive overview): plans focusing on one or two obstacles defined by the experimenter have been investigated [42–44] and observed inconsistent results. Bieleke et al. [34] argue that the effectiveness of if-then plans might be enhanced by tailoring if-then plans to athletes–that is, instructing exercisers to generate their own plan contents geared towards their personal obstacles and strategies to deal with them. For example, studies in which if-then plans were used without providing a particular obstacle in non-endurance domains (e.g., tennis, [45]; or golf and darts, [46]) found positive effects on performance. Thus, the heterogeneity of prior findings might be due to a lack of adequately tailoring if-then plans to the challenges an exerciser faces during the exercise [47].

### The present study

Taken together, the first aim of this study is to use a qualitative approach to investigate 1) the specific obstacles exercisers face during two different exercises, 2) the strategies they use spontaneously to deal with these obstacles, and 3) the strategies they consider helpful for future performances of the test. These obstacles and strategies were assessed after a first session with an anaerobic (Wingate test) and an aerobic cycling test (incremental exercise test). The second goal is to investigate whether if-then plans can help exercisers to better deal with the obstacles they face during an anaerobic and an aerobic cycling test. To this end, we used the obstacles and strategies that participants reported in the first session to help them generate individually tailored if-then plans for a second session with the same two tests. In line with previous research [45, 46], we expected improved performance in the if-then plan condition compared to a goal intention condition. As control variables, we assessed ratings of perceived exertion (RPE; [48]) and goal commitment in both sessions.

## Methods

### Participants and design

We recruited a sample of $N = 59$ participants (age: $M = 23.9 \pm 6.5$ years) for a 2-within (Session: 1 vs. 2) × 2-between (Condition: goal intention vs. implementation intention) mixed-factorial study. Study participants were recruited throughout the semester until a large enough sample was tested. The sample size was determined in accordance with other implementation intention research in sports, following recommendations to enhance power in this type of research (e.g., employing a within-subject design; for a systematic review, see [34]). A sample size of $N = 54$ participants is sufficient to detect a medium effect of $f = 0.25$ [49] in a mixed factorial ANOVA to find a within-between interaction ($\alpha = .05$) with a power of .95 (calculated with G*Power; [50]). Participants were randomly assigned to conditions. Seven participants were excluded from quantitative data analysis because they did not participate in the second session ($N = 3$), due to technical difficulties ($N = 3$), or because they did not comply with instructions ($N = 1$). This left 27 participants in the goal intention condition and 25 participants in the implementation intention condition.

The participants included in the quantitative analyses were $M = 169.4 \pm 16.1$ cm tall and weighed $M = 66.2 \pm 16.9$ kg. They reported to exercise $M = 5.4 \pm 2.9$ hours per week (endurance training: $M = 2.2 \pm 2.2$; strength training: $M = 2.4 \pm 2.2$) and to be engaged in a variety of sport activities (most frequently reported were jogging 21.1%, fitness 9.6%, soccer, weight training 7.7%), having performed their main sport activity for an average of $M = 7.3 \pm 7.0$ years. No participant reported cycling as their primary sport. Overall, participants stated to ride a bike for $M = 2.3 \pm 2$ hours per week as a physical activity (e.g., to commute to work), and to spend $M = 0.3 \pm 0.7$ hours per week with cycling as a sports activity. Participants in the goal and the implementation intention condition did not differ in their weekly training duration, $p = .370$, or the duration of performing their main sport, $p = .340$. There were no specific exclusion criteria for participation in the study, except that participants should have no injuries that prevented them from cycling. Moreover, we asked to avoid alcohol and strenuous exercise one day before each session and on the day of the session itself, as well as to refrain from consuming caffeine in the two hours before each session. Only marginal non-compliance from these requirements was recorded, with no differences between the goal and the implementation intention condition, $p$s $> .130$. All participants signed an informed consent and were compensated with 30 Euro and course credit when they completed both study sessions. The study protocol and measurements were approved by the Ethics Committee at the University of Konstanz (approval #24/2016). For all supplementary materials (Tables, Figs, study materials), please see https://osf.io/mq6kt/.

## Procedure

The study comprised two individual sessions in the laboratory that followed a very similar procedure (see S1 Fig). We kept the order of tests constant (i.e., the anaerobic test followed by the aerobic test) because for relatively untrained participants it would be very difficult to perform well in the anaerobic after completing the aerobic test. The second session took place 7 to 14 days after the first session at the same time as the first session (if possible for participants). Each session was carried out by a researcher who explained the study protocol and guided participants through the anaerobic and the aerobic test. To ensure standardization, the study was always conducted by the same two researchers who followed a prepared experimental protocol.

**First session.** After anthropometric measurements were taken, the ergometer settings were adjusted until participants were comfortable on the bike. Then, participants completed a standardized warm-up protocol (a three-minute ride with a resistance of 1.5 watt (W)/kg, a five-second sprint with a resistance of 10 W/kg, and a two-minute ride with a resistance of 1.5 W/kg), were given the opportunity to adjust their position on the bike, and were introduced to the procedure of the anaerobic test and asked to set a performance goal ("I want to reach maximum power as fast as possible and hold it for as long as possible."). To assess perceived exertion, participants reported their RPE on the category ratio 10 (CR10) scale [48, 51] prior to the test (to familiarize themselves with the use of the scale) and immediately after they completed it (a continuous measurement during the anaerobic test was not possible due to cycling at maximum power output). In the following fifteen-minute break, goal commitment was assessed, and a structured interview was conducted which explored perceived obstacles of participants, the strategies they spontaneously employed to deal with these obstacles, and the strategies they deemed helpful in future performances of the test (see study materials in S1 Appendix). After the break, participants were introduced to the aerobic test and asked to set a performance goal ("I want to cycle as long as possible."). During the test, RPE was repeatedly assessed every three minutes $\pm$ 15s (i.e., when resistance increased). Concluding Session 1,

another structured interview was performed to explore obstacle and strategies and participants completed a final questionnaire. During both tests, the researcher did not interact with participants and stayed outside their field of vision.

**Second *session*.** Except for the random assignment of participants to the goal intention or the implementation intention condition, Session 2 followed the same general procedure as Session 1. Participants in the goal intention condition wrote down their respective performance goals ("I want to reach maximum power as fast as possible and to hold it as long as possible." / "I want to cycle as long as possible."). They were also instructed to reflect on the situations, thoughts, and actions they had stated in the structured interview conducted in the first session, which were listed on an instruction sheet. Participants in the implementation intention completed the same steps but were afterwards instructed to explicitly link their stated obstacles with goal-directed actions or thoughts in an if-then format. These were individual if-then plans and thus varied between participants (e.g., "If I have some power left, then I will pedal once again with maximum force!"; "If exertion gets too high, then I will cheer myself on once more!"). Participants could formulate and write down as many if-then plans as they needed, then they were asked to reflect those if-then plans for the upcoming test. The formulation of several plans should allow participants to prepare effectively for different obstacles (cf. [45]).

## Tests and measures

**Performance tests.** Participants performed two different cycling tests that are frequently used in exercise science and which are typical in the context of endurance performance: the Wingate test (referred to as the anaerobic test; [52]) and an incremental exercise test (referred to as aerobic test). The Wingate test is a very short test (max. 30 seconds) that is used to measure peak power in a seated sprint on a cycling ergometer [53]. It is the most commonly used test when it comes to measuring anaerobic performance [54], is associated with a positive influence on endurance performance [55], and can also be used to predict endurance performance [56]. The incremental exercise test can be used to measure aerobic endurance performance [57]. Both cycling tests were performed on a Cyclus2 ergometer [58] that was equipped with Look Kéo Sprint clipless pedals.

*Anaerobic test.* Participants were instructed to cycle at a frequency of 70 revolutions per minute (rpm) and to accelerate then: At a frequency of 80 rpm, the program activated the anaerobic test which lasted 30 seconds. Resistance during the test was dependent on cadence and determined relative to participants' body weight [53]. Anaerobic exercise (maximum power) is generally reached after several seconds and decreases shortly (1–2 s) thereafter until the end of the test [59]. Participants were instructed to attempt to reach their maximum power after three to five seconds, and were asked to remain in a seated position, riding in the drops. As a measure of performance, the maximal performance in watts divided by bodyweight was used.

*Aerobic test.* Participants were instructed to cycle (seated and riding on the hoods) for as long as possible. The initial resistance was set to 60 W [60] and increased by 20 W every three minutes. Participants were instructed to keep a constant cadence between 75 and 95 rpm. On the Cyclus2 screen, participants could see the time until the next power increase, their cycled distance in kilometers, work in kilojoules, and their cadence. The test was terminated when participants' cadence dropped below 65 rpm, and they were prompted by the experimenter when the cadence came down to 70 rpm. Time-to-exhaustion was used as a measure of aerobic performance.

**Rating of Perceived Exertion (RPE).** To measure the effort associated with performance of the anaerobic and the aerobic test, we assessed participants' ratings of perceived exertion

(RPE) with the CR10 scale [48, 51]. RPE was described to participants as "the conscious sensation of how hard, heavy, and strenuous a physical task is" [8]. We printed a scale for RPE on a sheet of paper that was placed in front of participants. The scale ranged from 0 ("*nothing at all*") to 10 ("*maximal*") / 11 ("*even more than max*"; [61]).

**Questionnaires and structured interviews.** Participants stated their goal commitment (adapted goal commitment scale [62]; e.g., "*The goal was important to me.*"; four items in total) after both the anaerobic test (Session 1: Cronbach's α = 0.96, Session 2: Cronbach's α = 0.92) and the aerobic test (Session 1: Cronbach's α = 0.96, Session 2: Cronbach's α = 0.92) in both sessions on seven-point Likert scales (*1*: *does not apply*, *7*: *fully applies*). Additionally, participants provided demographic information (e.g., age, physical activity, main sports) once after the first session and they reported on their compliance with the study requirements (e.g., abstaining from caffeine) after both sessions.

In the structured interview, participants were asked to name any thoughts, emotions, and behaviors that they considered as obstacles for performing optimally in the test. Furthermore, they were asked to report any strategies they had used or that they deemed useful in future performances of the test. The experimenter explicitly stated that participants should focus on sensations and behaviors on which they have influence and that they could change. They were asked to answer as spontaneously as possible. Following these open questions, more detailed questions were asked by the experimenter (e.g., "Did a thought occur to you during the test, which you regard as helpful right now? When did this thought occur?" or "Were there any situations in which a hindering thought occupied your mind?"; analogous questions were asked regarding emotions and behavior). Only when participants struggled with answering these questions, examples were provided by the experimenter (e.g., "many athletes report encouraging themselves if the test becomes strenuous") and participants indicated whether these examples applied to them. If participants still did not address the question, the experimenters moved on to the next question.

## Data analysis

**Qualitative data.** Thematic analysis was used to analyze the structured interviews [63]. We used an inductive approach where the data were not assigned to any pre-determined categories and its level was semantic, i.e., it was not intended to identify underlying assumptions of participants [63]. One researcher first familiarized herself with the data to gain an initial impression of participants' individual obstacles and strategies. Then, all interview items were coded, meaning that they were grouped first into broad topics and then into more specific categories. These categories were then re-evaluated to make sure that they presented the dataset accurately. Items that related to more than one category were assigned to multiple categories. Concluding this first step of the qualitative analysis, each category was clearly defined and named. Then, the second step of the analysis followed, in which two independent raters categorized the interview items into the defined categories. A blind rating format was used, meaning that the two independent raters did not know the first coder's assignment and were only informed about the purpose of the interviews, the names of the categories and their definitions. If one item was coded to more than one category, raters were indicated to rate this item with multiple categories or to comment if they did not support multiple assignment. Then, the initial coder and the two raters discussed disagreements in assignment of categories until mutual agreement was achieved (see [64]) which was the case for all items. Categories were then grouped into general themes (see Table 1). S1A–S1C Table show the main results of the thematic analysis and give an overview over the categories, their definitions, and example statements, while Fig 1A–1C visualizes the frequencies of the general themes for both tests. S2

**Table 1. Overview over general themes and corresponding categories.**

| Obstacles | | Strategies used | | Potential strategies | |
|---|---|---|---|---|---|
| **General Theme** | **Category** | **General Theme** | **Category** | **General Themes** | **Category** |
| **Missing focus** | Distraction by screen* | **Distancing** | Cut out* | **Distancing** | Cut out thoughts |
| | Distraction through thoughts* | | Distraction** | | Distraction |
| | Distraction** | | | | Imagination |
| **Missing drive** | Incentive | **Attentional focus** | Screen | **Attentional focus** | Screen |
| | Demotivation | | Screen: Cadence* | | Screen: Cadence** |
| | Duration* | | Screen: Time* | | Screen: Time** |
| | Thoughts about stopping** | | Body* | | Body |
| | Performance reduction** | | Focus on test / goal* | | Concentration** |
| | Periods of time** | | Concentration** | | Technique** |
| | Screen** | | | | |
| **Negative sensations** | Frustration | | Technique** | | |
| | Shame* | | | **Drive** | Ambition |
| | Failure* | **Drive** | Ambition | | Attitude |
| | Arousal at Start / Finish* | | Attitude** | | Imagination |
| | Exhaustion* | | Imagination | | Motivation through nice thoughts* |
| | Exertion | | Motivation | | Rationalization |
| | Boredom** | | Rationalization* | | Self-Encouragement |
| | Pain** | | Self-Encouragement | | Screen** |
| | Pressure to perform** | | Joy | | Pride** |
| **Test demands** | Surprised by test demands* | | Pride | **Performance** | Technique* |
| **Cycling strategy** | Acceleration* | | Self-Worth** | | Orientation on screen: Cadence* |
| | Power management* | | Flow** | | Orientation on screen: time* |
| | Slowing down* | | Exertion | | Adjustment** |
| | Riding behavior / technique** | **Comfort** | Position** | | Take off pressure** |
| **Discomfort** | Ergometer* | **Miscellaneous** | Miscellaneous** | | Pressure to perform** |
| | Posture | **Planning** | Planning** | **Planning** | Planning* |
| | Body | **Pressure to performance** | Easiness* | **Goal** | Goal achievement* |
| **Goal** | Goal achievement** | | Sense of duty* | | Goal setting |
| | Aimlessness** | **Goal** | Goal setting** | | Goal focus** |
| | | | Goal focus** | | |

*Note.* * refers to obstacles / strategies that were only mentioned after the anaerobic test.

** references obstacles / strategies that were only mentioned after the aerobic test.

Table shows if- and then components (clustered in subcategories) used in the anaerobic and the aerobic test.

**Quantitative data.** First, we controlled if goal commitment in both tests and RPE finish in the anaerobic test differed between the goal intention and the implementation intention group and between sessions with a 2 (Condition: goal vs. implementation intention condition) × 2 (Session: 1 vs. 2) Bayesian mixed-factor ANOVA. For testing if RPE increased during the aerobic test and whether these increases differentiate between both conditions and sessions, we performed an analogous analysis with RPE as dependent variable, where we added Test duration as additional within-participants factor for the aerobic test (to account for differences in test duration in the aerobic test, RPE values were standardized from 0 to a 100 percent and then aggregated across 20% intervals).

In addition, we checked for any performance differences between both groups in Session 1 conducting Bayesian *t*-tests, for which we report Bayes factors ($BF_{10}$). The Bayes factor allows

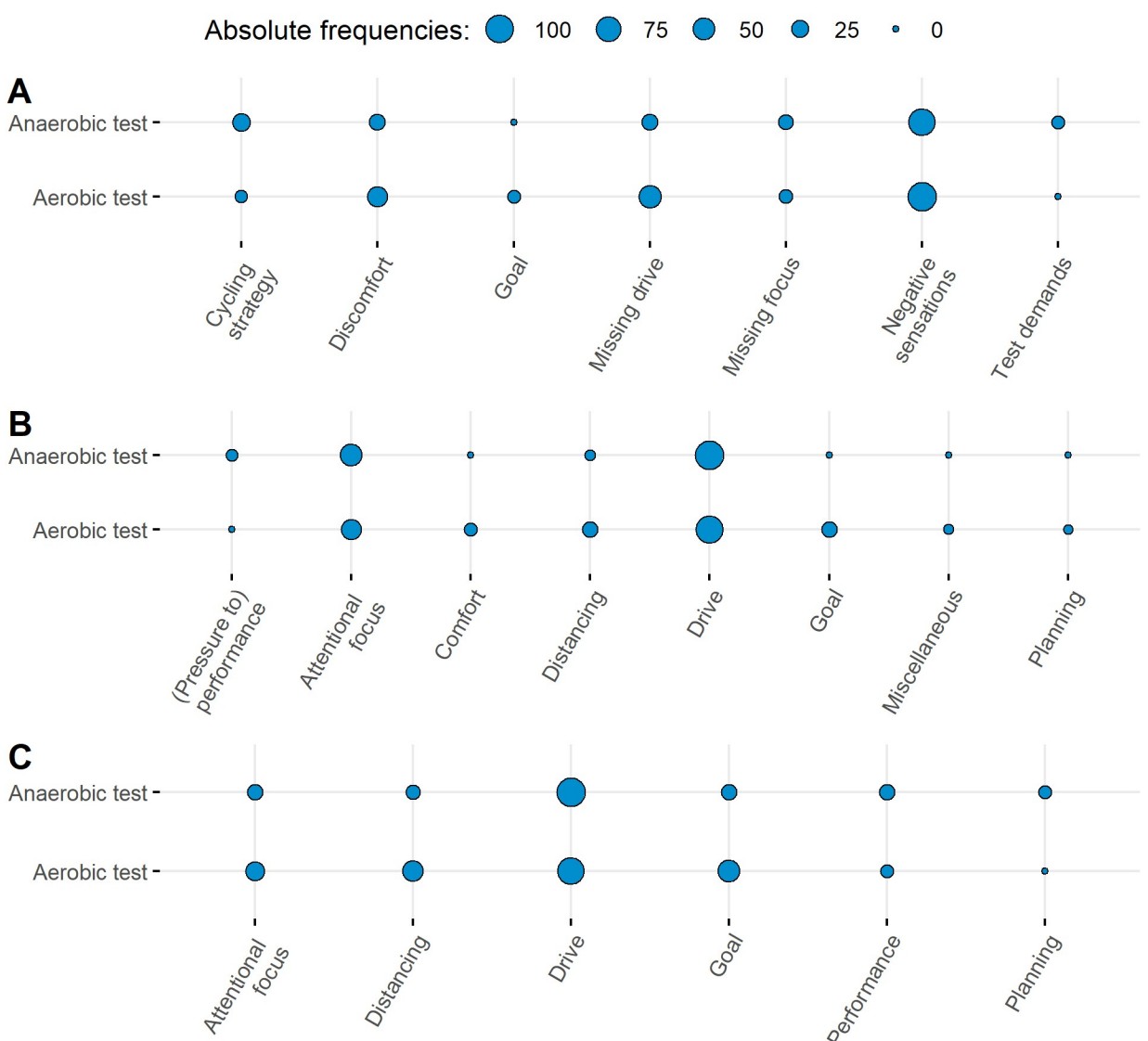

**Fig 1. Visualization of general themes and their frequencies of (a) experienced obstacles, (b) strategies used, and (c) potential strategies for overcoming experienced obstacles during the anaerobic and the aerobic test.**

a quantification of the relative predictive performance of the null hypothesis compared to an alternative hypothesis [65]. To aid the interpretation of these Bayes factors, we relied on the evidence categories suggested by Jeffreys [66] and Lee & Wagenmakers [67]. When data were not normally distributed, Bayesian Mann-Whitney-tests were used.

To test whether the implementation intention intervention improved performance in the second session compared to the first session, we ran 2 (Condition: goal vs. implementation intention condition) × 2 (Session: 1 vs. 2) Bayesian mixed-factor ANOVA. For anaerobic performance, peak power in watts divided by bodyweight was used as the dependent variable. Time-to-exhaustion was used as the dependent variable in the aerobic test.

Analyses were run in the statistical software environment R (3.3.6; [68]), using the Bayes-Factor package [69], and in JASP [70]. Plots were created using GGPLOT2 (3.2.1; [71]).

## Results

### Qualitative analysis: Obstacles, strategies, and if-then plans

The qualitative analysis covered the obstacles participants encountered during the anaerobic and the aerobic tests as well as the strategies they already used or that they considered useful for dealing with the obstacles in future test performances. As an overview, the resulting general themes and categories across tests are presented in Table 1. Fig 1A–1C provides an overview of the general themes and their frequencies, separately for each test. See S1A–S1C Table for the definitions of each category and examples of statements and S2 Table for if- and then components (clustered in subcategories) applied in the anaerobic and the aerobic test.

**Obstacles experienced during performance.** *Obstacles experienced in both the anaerobic and the aerobic test*. Effort was the most reported obstacle in both tests (e.g., "it's getting harder and harder"). Participants stated that frustration or anger about their own performance was obstructive (e.g., "frustrated when resistance was higher than expected"). Some participants reported discomfort regarding the ergometer, were occupied with their physical reactions (e.g., "getting nauseous"), or their position during the test, which they considered hindering. Demotivation to exert oneself was perceived as an obstacle as well. Some reported to have been distracted by information on the screen or by negative thoughts. Especially in the aerobic test, not being able to distract oneself was perceived as an obstacle. Some participants stated the test was not important to them and indicated a lacking incentive for the task.

*Specific obstacles in the anaerobic test*. The feeling of failure was reported to be hindering as well as shame during physical exercise. Specific periods of time of the test put some participants under pressure (e.g., "last 10 seconds"). As the test continued, exhaustion was an obstacle (e.g., "the urge to quit"). Further obstacles concerned difficulties with acceleration or to not having used maximum power during the test. Most of them regarded the moment when their cadence dropped as critical. Finally, some participants were surprised about resistance during the test, what led them to stop pedaling or to feel demotivated to keep going as goal achievement was harder than expected. Test duration was also perceived as demotivating.

*Specific obstacles in the aerobic test*. Feeling bored was reported to be hindering (e.g., "How long do I have to sit here") as was pain. Also, pressure to perform was perceived as aversive for some participants, especially when they compared themselves to others. Participants also reported riding technique to be a challenge or reported the information on the display to be demotivating. Being able to terminate the test anytime was reported as an obstacle, too. Specific time periods of the test (e.g., "always second half of the interval") demotivated participants as well as the perception of decreasing performance. Not being able to reach one's goal or not having an exact goal because the test had no fixed ending was also stated as obstacle.

**Strategies used to deal with obstacles.** *Strategies used in both the anaerobic and the aerobic test*. A majority of participants reported that motivation for the test in general was helpful during the test, especially motivation through self-encouraging statements (e.g., "give your best"). Also, participants perceived joy for exercise and exertion as helpful as well as the anticipation of being proud of oneself and of the own performance. Some stated to be motivated by feeling their own exertion (e.g., "exertion is progress") or by imagining cycling in other contexts. Many participants reported ambition to be helpful and motivating (e.g., "as fast as possible"). In general, concentration on information about the current ride presented on the screen was perceived as helpful.

*Specific strategies in the anaerobic test*. Rationalization of the test was helpful to keep performing (e.g., "I wanted to exercise more anyways"). Concentration on the screen was reported as beneficial as well, specifically regarding information about time and on cadence. Focusing on certain body parts or movements was also perceived as helpful as well as focusing on the

test. Few participants stated that it was beneficial to not put pressure on themselves (e.g., "it's ok to cycle like that") or that they felt obliged to follow the study protocol. Finally, some participants reported that it was helpful to cut out everything during the test.

*Specific strategies in the aerobic test.* Motivation through a positive attitude towards the test (e.g., "exercise makes you happy"), motivation through focusing on one's self-worth, and achieving a cycling flow (like a rhythm) were all perceived as helpful. It was helpful for participants to focus on themselves, on their technique (e.g., "pedaling constantly"), or to focus on the screen. Distraction was perceived as beneficial as well as focus on proximal goals or setting a goal. Few participants stated that it was helpful to adjust their posture during the test or to plan ahead. The category "Miscellaneous" contains items that were not applicable with the other categories.

**Potential strategies for dealing with the obstacles.** *Potential strategies in both the anaerobic and the aerobic test.* A lot of participants considered self-encouragement as being an effective behavior in critical situations, primarily through cheering themselves on. Also, ambition and a positive attitude was reported to be a potential strategy. Similar to the applied strategies, imagination of a motivational context (e.g., "doing a final sprint") or rationalization of the test was stated to be an effective thought. Specific goal setting was also reported as being potentially useful after both tests (e.g., "cycling at least 15 minutes, reaching a cadence of 75 one last time"). In general, concentration on the screen and on the body was reported. Finally, some participants stated distraction by thinking consciously of something else (e.g., "imagine song in my mind"), cutting out all thoughts, or imagining another environment or motivational things.

*Specific potential strategies in the anaerobic test.* Some participants reported that thoughts about something positive or goal achievement could be used to motivate them. Planning certain (tricky) parts of the test in advance was regarded as being potentially helpful (e.g., "control breathing, do not breathe hectically when resistance hits"). Optimizing performance through concentration on cadence or on time (e.g., "looking at the timeline, half is already done") on the screen was more prominent after the aerobic test, but also mentioned after the anaerobic test. Concentration on certain techniques was perceived as an effective measure to optimize performance.

*Specific potential strategies in the aerobic test.* Goal focus and feeling pride of oneself (e.g., "thinking about the success") was mentioned as potentially effective. Concentration in general, on cadence or on time, or on riding technique was proposed as an effective measure. Some participants stated that using screen information to adjust performance, taking off pressure (e.g., "cadence is alright, just as good as a faster one") and avoiding failing could be effective reactions to critical situations.

**Combining obstacles and strategies: Which if-then plans did exercisers form?.** In the second session, participants formed on average $M = 2.7 \pm 1.3$ ($Min = 1$, $Max = 5$) implementation intentions for the anaerobic test and $M = 3.1 \pm 1.5$ ($Min = 1$, $Max = 5$) for the aerobic test. About one third of the specified goal-related obstacles (if-components) pertained to exertion (e.g., "If I think about my tired legs. . ."), the second most specified if-component concerned the start/finish of the test (e.g., "If I reach the last 10 seconds of the test. . ."). Concerning the then-components, strategies comprising self-encouragement (e.g., ". . .then I'll smile and tell myself: everything's ok!") and ambition (e.g., ". . .then I'll stay ambitious and be better than the others!") were most frequently chosen. For the aerobic test, more than 40 percent referred to if-components concerning exertion (e.g., "If I nearly cannot go on any longer. . .") or certain periods of time during the test (e.g., "If the test begins. . ."), while around 36 percent of then-components targeted goal setting (e.g., ". . .then I'll finish the end of this step!") and distraction (e.g., ". . .then I'll distract myself by singing songs in my head!") as strategies.

## Quantitative analysis: If-then plans and performance

**Preliminary analyses.** *Goal commitment and RPE.* A comparison of participants in the goal and implementation intention condition regarding their goal commitment (see Table 2 for descriptive statistics) provided most support for the null model, suggesting no condition differences in both tests and in both sessions (i.e., 1 and 2), with all $BF_{10} \leq 0.45$ (anaerobic test) and all $BF_{10} \leq 0.61$ (aerobic test). The analysis of RPE finish in the anaerobic test indicated that our data provided most support for the null model, suggesting no differences in RPE finish between conditions and sessions, all $BF_{10} \leq 0.26$. Mean values of RPE finish indicate exertion after the aerobic test. As expected, RPE substantially increased in the aerobic test (all $BF_{10} \leq 0.26$).

*Performance in the first session.* Comparing the goal ($M = 8.3 \pm 0.8$) and the implementation intention condition ($M = 8.9 \pm 2.0$) in the anaerobic test at the first session (Fig 2A) provided little evidence for differences between conditions, $BF_{10} = 0.58$. In regard to the aerobic test (Fig 2B), we found slight evidence ($BF_{10} = 1.38$) for higher time-to-exhaustion in the implementation intention ($M = 24.0 \pm 7.5$) than in the goal intention condition ($M = 19.9 \pm 7.4$).

**Performance in the second session.** *Anaerobic test.* The analysis revealed that the data provided strongest support for a model with the main effect of Session, reflecting that performance improved from the first to the second session (see Table 4). In comparison, a model with the main effects of Session and Condition, $BF_{10} = 0.81$, and a model with both main effects and their interaction, $BF_{10} = 0.23$, received less support. This corresponds to slight and moderate evidence in favor of the Session main effect model, respectively. A model with the main effect of Condition received considerably less support, $BF_{10} = 5.52e^{-3}$, which constitutes extreme evidence in favor of the Session main effect model. Together, these findings suggest that the implementation intention intervention did not improve performance in the second session compared to the goal intention condition (for descriptive statistics, see Table 3).

*Aerobic test.* The analysis revealed that our data provided most support for the null model (see Table 4). In comparison, the model with the main effect of Condition received less

**Table 2. Descriptive statistics of goal commitment and RPE in the anaerobic / aerobic test for each condition (N = 25 for the goal intention condition (GI), N = 27 for the implementation intention condition (II).**

| Test | Measurement | Session | | *M* | *SD* | *95%CI* |
|---|---|---|---|---|---|---|
| **Anaerobic** | Goal commitment | 1 | GI | 5.9 | 1.2 | [5.4, 6.4] |
| | | | II | 5.8 | 1.6 | [5.2, 6.5] |
| | | 2 | GI | 5.6 | 1.4 | [5.1, 6.2] |
| | | | II | 6.3 | 0.6 | [6.0, 6.6] |
| | RPE finish | 1 | GI | 7.0 | 1.6 | [6.4, 7.7] |
| | | | II | 6.6 | 1.5 | [6.0, 7.2] |
| | | 2 | GI | 7.0 | 2.2 | [6.1, 7.9] |
| | | | II | 7.1 | 1.7 | [6.4, 7.8] |
| **Aerobic** | Goal commitment | 1 | GI | 6.0 | 1.3 | [5.5, 6.6] |
| | | | II | 6.0 | 1.6 | [5.4, 6.6] |
| | | 2 | GI | 6.0 | 1.4 | [5.4, 6.6] |
| | | | II | 6.5 | 0.5 | [6.4, 6.7] |
| | RPE | 1 | GI | 5.1 | 0.9 | [4.7, 5.5] |
| | | | II | 5.2 | 0.7 | [4.9, 5.5] |
| | | 2 | GI | 5.4 | 0.8 | [5.0, 5.7] |
| | | | II | 5.1 | 0.7 | [4.8, 5.4] |

RPE in the anaerobic test refers to RPE at finish, RPE in the aerobic test is a mean value of reported RPE).

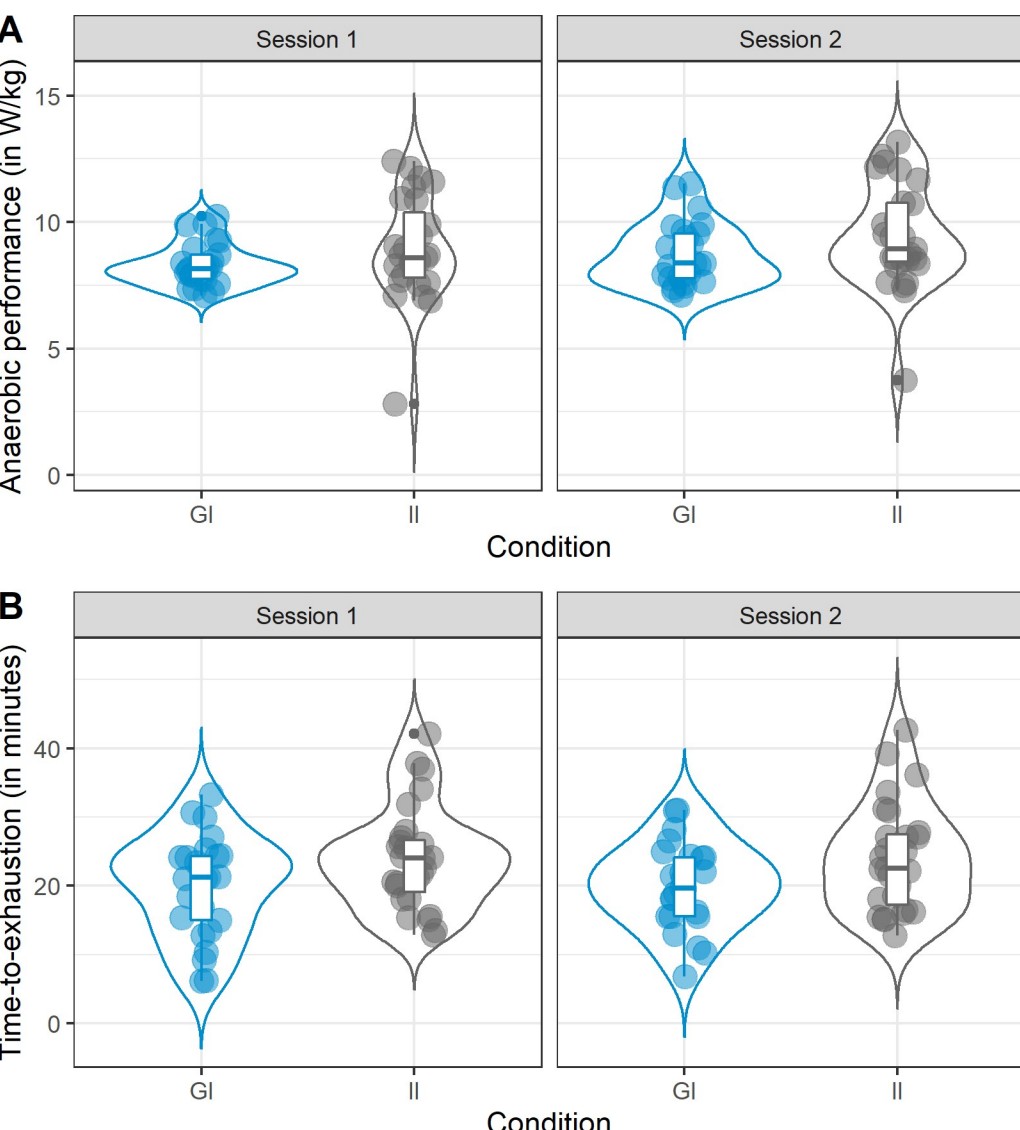

**Fig 2. Violin plots and boxplots of (a) anaerobic performance in the anaerobic test and (b) time-to-exhaustion in the aerobic test as a function of Condition (goal vs. implementation intention).**

support, $BF_{10} = 0.62$, which corresponds to little evidence in favor of the null model. The model with the main effect of Session, $BF_{10} = 0.20$, and the model with both main effects, $BF_{10} = 0.16$, received less support as well. This corresponds to moderate evidence in favor of the null model. Finally, the model with both main effects and their interaction received considerably less support, $BF_{10} = 0.04$, suggesting strong evidence in favor of the null model. Together, these findings suggest that time-to-exhaustion in the aerobic test was not influenced by Session (i.e., no learning effects) or Condition (i.e., no intervention effects).

## Discussion

We assessed the obstacles exercisers with little or no prior cycling training experience faced during two different cycling tests, along with the strategies they spontaneously applied to deal with these obstacles and strategies they considered helpful for future performances of the tests.

**Table 3. Summary of means, standard deviations and 95% confidence intervals for anaerobic power (in W/kg) and mean power (in W) in the anaerobic test / time-to-exhaustion (in minutes) and maximum power (in W) in the aerobic test for each condition (N = 25 for the goal intention condition (GI), N = 27 for the implementation intention condition (II)).**

| Test | Measurement | Session | | *M* | *SD* | *95%CI* |
|---|---|---|---|---|---|---|
| **Anaerobic** | Anaerobic power (in W/kg) | 1 | GI | 8.3 | 0.8 | [8.0, 8.7] |
| | | | II | 8.9 | 2.0 | [8.1, 9.7] |
| | | 2 | GI | 8.7 | 1.2 | [8.2, 9.2] |
| | | | II | 9.4 | 2.0 | [8.6, 10.3] |
| | Mean power (in W) | 1 | GI | 405.6 | 97.8 | [365.2, 445.9] |
| | | | II | 484.6 | 137.7 | [430.2, 539.1] |
| | | 2 | GI | 412.4 | 87.0 | [376.5, 448.3] |
| | | | II | 495.6 | 135.3 | [442.0, 549.0] |
| **Aerobic** | Time-to-exhaustion (in minutes) | 1 | GI | 19.9 | 7.4 | [16.9, 23.0] |
| | | | II | 24.0 | 7.5 | [21.1, 27.0] |
| | | 2 | GI | 20.0 | 6.2 | [17.4, 22.5] |
| | | | II | 24.0 | 7.8 | [21.0, 27.2] |
| | Maximum power (in W) | 1 | GI | 188.0 | 50.3 | [167.2, 208.8] |
| | | | II | 211.8 | 47.8 | [192.9, 230.8] |
| | | 2 | GI | 187.2 | 41.6 | [170.0, 204.4] |
| | | | II | 215.6 | 53.0 | [194.6, 236.5] |

Building on this qualitative approach, we then tested whether combining the perceived obstacles and the identified strategies into self-regulatory if-then plans helped exercisers improve their performance in a second exercise session.

## Experienced obstacles, applied strategies, and potential strategies

For both tests, participants identified multiple exercise-induced obstacles (Fig 1A), spontaneously applied a diverse range of strategies (Fig 1B) and considered several strategies helpful for

**Table 4. Model comparisons with Bayesian mixed factor ANOVA: Dependent variables were anaerobic power (maximal performance in watts divided by weight) for analysis of performance in the anaerobic test / time-to-exhaustion (in minutes) for analysis of performance in the aerobic test.** Each model was compared against the best model.

**Anaerobic test**

| Models | P(M) | P(M\|data) | $BF_M$ | $BF_{10}$ | error % |
|---|---|---|---|---|---|
| Session | 0.20 | 0.49 | 3.81 | 1.00 | |
| Session + Condition | 0.20 | 0.40 | 2.63 | 0.81 | 5.39 |
| Session + Condition + Session * Condition | 0.20 | 0.11 | 0.50 | 0.23 | 4.60 |
| Null model (incl. subject) | 0.20 | 2.70e -3 | 0.01 | 5.53e -3 | 2.02 |
| Condition | 0.20 | 2.69e -3 | 0.01 | 5.52e -3 | 3.13 |

**Aerobic test**

| Models | P(M) | P(M\|data) | $BF_M$ | $BF_{10}$ | error % |
|---|---|---|---|---|---|
| Null model (incl. subject) | 0.20 | 0.49 | 3.89 | 1.00 | |
| Condition | 0.20 | 0.31 | 1.77 | 0.62 | 5.60 |
| Session | 0.20 | 0.10 | 0.45 | 0.20 | 1.03 |
| Session + Condition | 0.20 | 0.08 | 0.34 | 0.16 | 8.71 |
| Session + Condition + Session * Condition | 0.20 | 0.02 | 0.08 | 0.04 | 8.28 |

*Note.* All models include subject. Prior and posterior model probabilities are depicted in the column P(M) and P(M|data), respectively. $BF_M$ illustrates the change from prior to posterior model odds, $BF_{10}$ the Bayes factor for each model, and Error % the precision of the Bayes factor calculations.

future performance of the tests (Fig 1C). Regarding obstacles, participants in the implementation intention condition most often specified perceived exertion, pressure-related thoughts at the start and finish, and distraction by information on the screen in the if-part of their plans in preparation for the anaerobic test. Regarding strategies, the most frequently used then-parts referred to self-encouragement, ambition, and planning. Interestingly, plan content was different in the aerobic test. Here, the obstacles specified in the if-part mostly referred to exertion, demotivation at critical moments, and demotivation in general. The goal-directed behaviors specified in the then-parts mainly referred to seeking distraction or focusing on riding technique. Accordingly, participants recognized and considered differences in the specific demands posed by the aerobic and anaerobic test in their if-then plans.

The emphasis on exercise-related sensations like exertion, pain, discomfort, or the urge to stop as obstacles replicates the findings of McCormick et al. [1]. Likewise, the difficulties to remain concentrated, to be negatively affected by distractions, and to respond with frustration to stressors during the test is in line with this prior research. In general, the identified obstacles and strategies during performance fit well into and further expand existing frameworks, like the distinction between associative and dissociative cognitions [72]. Interestingly, while McCormick et al. [1] interviewed endurance athletes, only a small portion of our sample were active in endurance sports ($n$ = 16) and even these athletes had negligible experience with cycling tests. This attests to the universality of the sensations associated with the performance of endurance sports. The importance of perceived exertion as a major performance-limiting obstacle aligns well with predictions of the psychobiological model [6, 7], supporting the crucial role perceived exertion plays for endurance performance. Furthermore, the relevance of reported exertion, pain, and discomfort seems to be characteristic of less experienced samples (e.g., [13]) and it can be assumed that a majority of our sample was overwhelmed by the external demands of the task (e.g., riding on an ergometer, staying in a seated cycling position) and their physical challenges (e.g., heat stress in the laboratory, muscle pain) that their concentration on the performance test objectives was limited. Especially during the anaerobic test, technical aspects of cycling like acceleration, power management, and slowing down were reported as critical by participants, which converges with findings that less experienced athletes experience greater difficulty in regulating their efforts [73] and in dealing with their physiological sensations compared to more experienced athletes [13]. In practice, it might be helpful to educate less experienced athletes about possible physiological sensations in advance and to suggest several effective then-strategies for dealing with common if-obstacles (e.g., effort, muscle pain) in order to create a helpful if-then plan (as effectively applied in other if-then planning studies in sport [45]).

Besides corroborating previous research about obstacles in endurance sports, the participants in our study also identified obstacles that have so far received little attention in sport psychological research. Most notably, a consistently stated obstacle in the aerobic test was boredom, which also was the fourth most frequently used if-component in the implementation intention condition. This suggests that participants not only got bored frequently during the aerobic test and attached great importance to boredom as an obstacle, but also searched for ways to deal with it. This is particularly important as boredom has long been overlooked as a potential obstacle in endurance sports [74]: Recent evidence points towards the relevance of boredom in elite [75] and recreational sports alike [76]. Considering boredom as an obstacle for endurance performance also aligns well with theoretical considerations, as recent work has highlighted that boredom can directly act as a self-regulatory challenge to goal pursuit [77], thereby making it harder for an athlete to perform optimally.

In contrast to prior research that took a broader view on the challenges of endurance sports (e.g., comprising competitions and training; [1]), we decided to take a more narrow view by

focusing on the obstacles faced during two very specific cycling performances (i.e., an anaerobic and an aerobic tests). In doing so, we answered calls for an in-depth exploration of the demands associated with specific performances [1]. This allowed us to present a detailed analysis of exercise-induced obstacles (e.g., identifying boredom as an obstacle in the aerobic test but not in the anaerobic test), which can be used as a starting point for developing more tailored sport-psychological trainings.

The strategies reported by participants in our study also align well with the existing literature. Particularly, the frequent use of self-encouraging statements seems to be an adaptive choice, as research has shown that self-talk can be beneficial for performance [78]. In addition, many of the reported strategies can be understood as emotion-regulation strategies, which also seems to be adaptive because such strategies can enhance positive emotions [79]. For example, participants found it helpful to imagine the feeling of being done with the test, to remind themselves of being able to finish the test, or to be proud about having achieved half of the test because of their own ambitions.

## The (lack of) efficacy of self-generated if-then plans

We found that performance was not enhanced by the use of if-then plans, neither in the aerobic nor in the anaerobic test. Prior research has shown that improving endurance performance with ready-made self-regulatory strategies that focus on one or two pre-defined obstacles is unlikely [42–44, 47]. Therefore, we instructed participants to generate individual plans, capitalizing on the information about obstacles and strategies we had elicited in structured interviews after the first session. Similar approaches are known to be effective in other domains; for example, when making if-then plans is combined with the self-regulation strategy of mental contrasting (i.e., mental contrasting with implementation intention, MCII; [80, 81]). An MCII intervention has parallels to our approach because it emphasizes the need to elicit desires and goals and to contrast them with the obstacles of attaining these goals. MCII supports has been shown to facilitate short-term and long-term goal pursuit attainment across various domains of life, such as academic accomplishments [82], relationships [83], health [84], and physical activity [85]. MCII is also known to be effective in the domain of sports: In a study investigating the use of MCII among dance sport athletes, the use of MCII was related to better performance [86]. It thus seems worthwhile to examine whether an MCII procedure would be more effective in enhancing endurance performance than an if-then planning intervention.

## Limitations

One drawback of relying on self-generated if-then plans is that researchers have little control over the plans participants specify. Consequently, it is possible that participants generate plans that do not affect their performance or even have detrimental effects. For instance, in one study on improving volleyball service performance [87] if-then plans were generated based on coaches' feedback regarding the performance of their athletes. The authors of this study argue that the resulting if-then plans might have directed attention too much on the execution of well-learned motor behaviors, which might have interfered with performance. It is conceivable that participants in our study also planned behaviors that they thought would help them perform better, while in fact these behaviors might have been ineffective or even detrimental in some cases. In addition, critical situations were sometimes specified in vague terms (e.g., "If it becomes exhausting") or did not pertain to an obstacle or opportunity (e.g., "If I am cycling"). This might be due to a lack of experience with cycling training, which required participants to base the specified obstacles and strategies on a single test performance. Consequently, it might be that if-then planning is a more effective strategy among experienced athletes (see [34], for a

similar argument). With this assumption in mind, the results of the qualitative analysis must be considered carefully. A generalization to more (e.g., athletes) or less (e.g., physically inactive) experienced populations can only be made to a limited extent. Future research should investigate the extent to which these findings hold in other populations as well.

A closer look on our qualitative findings provides additional insights into the question of why if-then plans might have been ineffective in our study. In both tests, participants were able to identify obstacles and spontaneously applied several strategies that have shown to be useful by prior research (e.g., emotional regulation, attention regulation). This alone might have helped participants in the goal intention condition to improve their performance. Tentative support for this reasoning comes from the observation that participants substantially improved their performance in the aerobic test between the first and the second session, irrespective of whether they belonged to the goal or the implementation intention condition. Thus, participants might have been very effective in finding ways to deal with the challenges posed by the tests, making it difficult for the if-then planning intervention to further improve performance. Similar effects have been observed in a study investigating the influence of self-talk on performance in a time-to-exhaustion ergometer test, in which the expected effects of self-talk were masked by learning effects through task repetition and concomitant reflection of participants [88]. Subsequent studies might resort to a less rigorous control condition in which participants do not engage in interviews about obstacles and strategies. Nevertheless, comparing an implementation intention condition with a goal intention condition is a standard approach in if-then planning research [35] and differences between conditions are commonly observed despite the possibility that participants in the goal condition might form spontaneous if-then plans. Furthermore, plan effects in future studies might be enhanced by asking subjects to make one plan pertaining to the most critical obstacle (or opportunity) and the best goal-directed strategy. Rather than helping subjects prepare for different stressors in the cycling tasks, there might have been interference between the different if-then plans [89]. Indeed, the majority of subjects in the implementation intention condition used more than one plan to prepare for the tests, which might have thwarted the beneficial effects of if-then planning. Finally, it is possible that the type of goals that participants set (e.g., process-related: keep going, ignore pain; outcome-related: target time) might have had an influence on endurance performance (see [90] about recommendations on effective goal setting in sports). We did not address this issue here because our focus was primarily on if-then planning rather than on goal setting. Still, future research should focus on how various types of exercise-related goals affect endurance performance. Also, future studies should consider the possible interaction of (un-) specificity of goals and if-then planning, as studies indicate that specific goals lead to better performance than vague, general or no goals [91] and goal specificity alone trumps the effects of merely 'do your best' goals [92]. It is plausible that forming specific goals already improve endurance performance and thereby reduce implementation intention effects.

Another reason for the lack of efficacy of our intervention could pertain to the characteristics of the sample. None of our participants performed cycling as their main sport, meaning that they had little to no experience with the specific task demands. Research indicates that there could be differences between participants who regularly engage in a certain activity compared to inexperienced participants in terms of their motivation to perform well [93]. With regard to sport psychological interventions, it has been argued that inexperienced samples lack the intrinsic motivation to perform the task, therefore obscuring any effects of self-regulatory strategies that are based on sufficient motivation [3]. The results of our qualitative analysis mirror this argument, as many participants reported missing drive and motivation as obstacles for performing well. However, our participants reported to be well committed in both conditions. Nevertheless, these results should be interpreted with care because they might reflect

social desirable responding. This underlines the necessity to conduct future experiments testing if-then planning in endurance sports with experienced athletes who are presumably much more intrinsically motivated to perform as good as possible and master the metacognitive skills and attentional strategies necessary to implement strategies during performance [4]. Furthermore, given the lack of task-specific cycling experience in our sample, it might have been especially difficult to apply if-then planning in an immediately following test: the effective use of psychological skills requires people to try whether the corresponding strategy is suitable for them [94]. This calibration time constitutes one reason for the recommendation to not make use interventions shortly before an event (like the performance tests for the participants; [3]). Future research could benefit from probing the efficacy of self-generated if-then plans prior to a test and to adapt the plans if necessary.

In addition, there are a number of factors that potentially determine endurance performance but are beyond the scope of the present investigation. For example, future studies should take the influence of self-efficacy into account, a widely studied determinant of sports performance (e.g., [95]) with beneficial effects on effort perception [96] and pain tolerance [97]. Moreover, high self-efficacy is associated with improved implementation intention effects in complex tasks [98], suggesting that it might be crucial to ensure that participants feel confident with performing the athletic task to reap the potential benefits of forming if-then plans. In the same vein, it can be discussed whether a different performance test could have been more appropriate to show effects of if-then plans. As the aerobic test requires a steady increase in power output and thus poses increasing self-regulatory demands, a test that feels more achievable (e.g., a time trial) could have heightened participants' perceptions of self-efficacy. However, as a self-paced time trial would induce further self-regulatory demands (e.g., constantly having to adjust one's power output to what one believes one can sustain for the expected ride time), this would be a less pure measure of aerobic performance–as pacing errors can substantially affect overall performance—compared to the aerobic test. A difference in mean power between conditions in Session 1 makes the elicitation of if-then planning effects even more intricate. On the one hand, as participants in the implementation intention condition already presented higher power output, additional enhancement of performance might have been difficult which could obscure potential if-then planning effects. On the other hand, performance parameters do not indicate that further improvement through if-then planning is not possible, which again raises the question of the appropriateness of the tests to illustrate performance enhancements. Future studies should investigate which tests are optimally suited to reflect if-then planning effects on endurance performance. Additionally, conducting both performance tests on the same day is a limitation of this study from an exercise physiology and psychology perspective: Perceptual and physiological processes in the first test might have influenced those in the second test (e.g., in terms of residual fatigue; see [99] for a detailed review).

## Conclusions

In this study, we investigated the obstacles exercisers face during an anaerobic and an aerobic cycling test, as well as the strategies they considered helpful for dealing with these obstacles. We further investigated whether if-then plans based on these obstacles and strategies lead to improved performance. Qualitative analyses of interviews conducted with participants immediately after the tests revealed substantial insights into diverse sets of obstacles and strategies. The quantitative analyses suggest that an if-then planning intervention did not improve anaerobic performance or time-to-exhaustion. These findings indicate that participants were able to identify exercise-related obstacles and useful strategies; moreover, performance improvements

in the aerobic test suggest that they could use this information to their advantage. However, if-then planning provided no additional benefits. When using if-then planning for less experienced athletes in practice, it might be helpful to educate exercisers about potential obstacles (e.g., muscle soreness) and to suggest potential strategies (e.g., self-encouragement), to use one if-then plan that is rehearsed with a sufficient time lag before use, and to ensure that exercisers are familiar with the sport so that they feel more self-efficient per se. Future research should complement our study with a focus on more experienced athletes and other intervention techniques. Nevertheless, our findings shed novel light onto the complex interplay of performance-related factors when investigating the impact of psychological interventions designed to help athletes in dealing with determinants of endurance performance.

## Supporting information

**S1 Table.** A. Obstacles the participants reported to have experienced during the anaerobic / aerobic test. Note. * refers to obstacles that were only mentioned after the anaerobic test, while ** references obstacles that were only mentioned after the aerobic test. B. Strategies used (thoughts, sensations or behaviors) during the anaerobic / aerobic test. Note. * refers to strategies that were only mentioned after the anaerobic test, while ** references strategies that were only mentioned after the aerobic test. C. Potential strategies reported by participants after the anaerobic / aerobic test. Note. * refers to potential strategies that were only mentioned after the anaerobic test, while ** references potential strategies that were only mentioned after the aerobic test.
(DOCX)

**S2 Table. If- and then-components in categories (incl. frequency) used by participants to enhance performance in the anaerobic / aerobic test.**
(DOCX)

**S1 Fig. Flowchart, visualizing the protocol of session one (a) and session two (b).** Manipulation means the random assignment to either the goal intention / implementation intention. condition.
(DOCX)

**S1 Appendix. Study materials.**
(DOCX)

## Acknowledgments

We thank Theresa Bäumle and Julia Heck for their assistance in data collection.

## Author Contributions

**Conceptualization:** Maik Bieleke, Raphael Bertschinger, Julia Schüler, Wanja Wolff.

**Data curation:** Anna Hirsch.

**Formal analysis:** Anna Hirsch.

**Funding acquisition:** Maik Bieleke, Raphael Bertschinger, Wanja Wolff.

**Methodology:** Maik Bieleke.

**Project administration:** Maik Bieleke, Julia Schüler, Wanja Wolff.

**Resources:** Maik Bieleke, Julia Schüler.

**Supervision:** Maik Bieleke, Julia Schüler, Wanja Wolff.

**Visualization:** Anna Hirsch.

**Writing – original draft:** Anna Hirsch.

**Writing – review & editing:** Anna Hirsch, Maik Bieleke, Raphael Bertschinger, Julia Schüler, Wanja Wolff.

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
