## [Decision Letter · Decision Letter 0]

10 Jun 2021

PONE-D-21-13120

Struggles and Strategies in Anaerobic and Aerobic Cycling Tests: A Mixed-method Approach with a Focus on Tailored Self-Regulation Strategies

PLOS ONE

Dear Dr. Anna Hirsch,

Thank you for submitting your manuscript to PLOS ONE. After careful consideration, we feel that it has merit but does not fully meet PLOS ONE’s publication criteria as it currently stands. Therefore, we invite you to submit a revised version of the manuscript that addresses the points raised during the review process.

The reviewers have now submitted their independent reports. They both agree that your study investigates an important topic and has several strengths. They however also raised several major concerns that must be addressed during the revision process. The issues raised concern both the style and the content. Please take particular attention to the importance of restructuring and shortening the introduction, as I agree with reviewer 2 that it is quite hard to follow in the present form. You must also justify the use of some measures and provide more details about the instructions provided to the participants. This is for instance the case for RPE scores, as it is unclear to what effort scores collected "at rest" (before the realization of an effort) are referring to. I anticipate a confusion between residual fatigue and effort, as it is frequent to collect residual muscle fatigue scores to ensure that a participant start an exercise trial in "fresh" conditions. The report of clear instructions may help resolving this issue.

I hope that the points raised in the reports will be helpful to you.

We look forward to receiving your revised manuscript.

Kind regards,

Mathieu Gruet, Ph.D

Academic Editor

PLOS ONE

Journal Requirements:

Reviewers' comments:

Reviewer's Responses to Questions

**Comments to the Author**

1. Is the manuscript technically sound, and do the data support the conclusions?

Reviewer #1: Partly

Reviewer #2: Partly

2. Has the statistical analysis been performed appropriately and rigorously? 

Reviewer #1: Yes

Reviewer #2: I Don't Know

3. Have the authors made all data underlying the findings in their manuscript fully available?

Reviewer #1: Yes

Reviewer #2: Yes

4. Is the manuscript presented in an intelligible fashion and written in standard English?

Reviewer #1: Yes

Reviewer #2: Yes

5. Review Comments to the Author

Reviewer #1: There is much to like about this paper and the effort that has gone in to the work. I believe that it is important to support publishing studies that have non-significant findings, but in this instance there are concerns that need addressing to help explain these non significant findings much more strongly. These relate to the design (no passive control condition, where participants did not reflect on their thoughts, selection of endurance tasks), the qualitative analysis, as well as some of the interpretations and discussion points. I have outlined my comments below. I appreciate there are a few of them, but I do hope that the authors find them useful in helping to improve the manuscript.

Abstract:

Suggest to rephrase ‘fine-grained investigations’. I think I understand the essence of it, but would suggest to adjust word choice as for some of the demands there are in depth investigations (i.e. mental fatigue), although I agree that the psychology of endurance performance has received relatively little research attention compared to some other sporting domains. Overall, the abstract can be written more clearly to capture the aim of the study better, as well as what was done (i.e. whether the interviews informed the if-then plan of the second visit).

Introduction:

1. On the first pages the authors present an overview of the exercise-related psychological obstacles that might affect performance. Although some relevant studies are reviewed I am not entirely clear on what the author’s interpretation is of what comprises obstacles. That is, some of the obstacles that are presented reflect strategies (active distraction, coping with pain). To help improve this section I would encourage the authors to be clearer on what is meant with obstacles and strategies (Birrer & Morgan, 2010 might be a useful resource here) and to provide a more coherent overview of the literature. For example, a study that is missing when discussing strategies in cycling is Blanchfield, Hardy, De Morree, Stainao, & Marcora’s (2014) study on self-talk.

2. Line 74: I agree that there it is a dearth of research that has specifically explored the mechanisms underpinning the potential success of an intervention in relation to endurance performance (i.e. see McCormick et al., 2015, for a discussion), but there are some think aloud studies that have explored obstacles (i.e. Whitehead, Jones, Williams, Dowling et al., 2019). Overall, I don’t think that the introduction has given justice to some of the research on endurance psychology that is out there.

3. An interesting point is raised in relation to the different physical and physiological demands of cycling (i.e. short prints and uphill cycling) and that these may require different psychological strategies. Most sports have different physiological demands (think of football as an example), so I would not think this is unique to cycling. Furthermore, although this may not have been tested directly in the cycling research, there are examples in the wider sport psychology literature. I would suggest to shift the focus on what evidence there is to suggest that tailoring one’s psychological strategy use to the specific demands of a task is helpful. This will help to show how this research builds on what is known in terms of the efficacy of interventions.

4. A clear overview of the relevance of if-then planning is provided. From line 131-137: Could the authors specify what the effects of different types of goals are (i.e. process goals (keep going) or outcome/performance goals (time-based or based on outcome of event such as medals)? This is quite relevant considering the central role of goals in the study design.

5. Line 150-151: To what extent do the author’s believe that self-efficacy is a key mechanism here?

6. The aims of the study are quite ambitious, and this has seemingly weakened the focus of the review of the literature and the research overall.

Specific comments:

Line 46: Suggest to replace marathon with ride to avoid confusion with running events.

Line 49-50: Although there is a limited research on the psychological demands of endurance performance, I would be careful with the phrasing of that it is poorly understood.

Line 57: What is meant with the limits of endurance performance? Clarification of this at the start of the paper would help to set the scene more clearly.

Method/Results:

1. A reference was made to statistical power – was a power analysis conducted? Ensure to include this.

2. Seven participants dropped out, did the authors look at the data of these participants? Did those who dropped out still get course credit and the monetary compensation? There might be some insightful information here from the interviews as well. For example if they (participants who dropped out) all found it very hard and not enjoyable and there is a pattern, then this is worthwhile reporting.

3. Participants: Would be insightful to have a greater idea of the sport activities the participants normally do – cycling was not anyone’s primary sport, yet on average they cycled a few hours a week, so it is a sport they are familiar with? How can they then be classified as exercisers with little or no prior cycling experience? I am also a bit confused about the hours per week that are reported, it is stated that the average exercise endurance training per week was 2.3hrs, which is the same as the number reported for cycling, but the SD is different? What happened here? Also, some greater clarity about the inclusion criteria would be useful.

4. As an observation: The performance goal examples were not very specific – i.e. go for as fast as possible, or hold it for as long as possible. If they have not already done so, the authors may want to consider looking at some of the ‘do your best’ goal research (see Burton and colleagues) and the impact of the specificity of goals. If goals are vague, then it can be more difficult to commit to these, and this is something they can return to in the analysis of the if-then plans used in the results.

5. Line 229: What was the justification of measuring RPE immediately before the test and what was the average at this stage? How quick was RPE taken after the test, and what were the instructions?

6. There are some issues with the conditions in terms that the participants in the goal intention condition were asked to reflect on the situations, thoughts, and actions they had stated in the structured interview. This could have (unintentionally) led to them setting spontaneous if-then plans and reflect on how they could do better. Did the authors consider this, and if so, how was this accounted for in the data analysis? Another issue is that the examples of ifs are quite vague or not necessarily a critical situation, i.e. if I have some power left.

7. Line 248: What does the literature say in terms of an ‘optimum’ number of if-then plans? If someone sets as many if-then plans is that a good thing? Would be useful to clarify this for the reader.

8. Line 264: Out if interest, by showing the participants feedback such as the time until the next power increase and so on, this can be a factor that can influence one’s coping. For example, if you know it is only a few seconds until the next step, then you keep going. They may have set themselves specific goals that influenced this. Is this something that was explored in the interviews and the data analysis?

9. Line 303-304: How was this re-evaluation done and the accuracy assessed? Is this feasible and does this align with the principles of conducting thematic analysis. Interrater reliability is generally not advocated as part of thematic analysis and is surrounded with controversy. Having said this, it may work for the current study, but perhaps better as a content analysis, do consider the suggestions by O’Connor & Joffe (2020) for example: https://journals.sagepub.com/doi/full/10.1177/1609406919899220 A major issue with this type of coding is that one participant may have been quite in depth in their answers and as such they could have given multiple references to (as an example) motivation or frustration in their answers. What seems to have happened in the analysis (but I may have misunderstood) is that this was not accounted for, and as such the use of percentages in table S2x is meaningless.

10. Looking at the data, it seems that missing drive and motivation was a major obstacle for many of the participants. This may well have affected a lot of the results and the effectiveness of the plans.

11. A nice insight was provided into the content of the if-then plans, and this is valuable data to share.

12. Perhaps unfortunately, there seemed to be a difference in terms of power for both tests between the two conditions at the first session which is difficult to control for at the start. Nevertheless it is interesting that, for the aerobic test the time to failure between the two conditions was pretty much identical. Perhaps this tells us more about the appropriateness of the test, rather than the effectiveness of if-then plans. A discussion of this and whether a 20 minute time-trial (for example) would have been more suitable.

Discussion:

A few comments that I have made earlier could be considered in the discussion, such as the appropriateness of the cycling task used (for example, why not a self-paced task, or a 20 min time trial), drop out of participants, the alternative condition (goal condition) chosen, discussing of mechanisms that can make if-then plans effective (perhaps self-efficacy) that could have been measured? A combination of the latter two points is relevant as the type of tasks (regularly used as indicators of fitness and peak power) may have prevented the authors to identify a ‘performance’ change but the participants may have felt more confident. I am not surprised that there was no effect on performance. Thus, I would not necessarily support the conclusion that there is a lack of effectiveness of the if-then plans, because effectiveness was not actually considered beyond a performance measure of power (arguably this focused more on efficacy than effectiveness? See Bishop, 2008 for a discussion of effectiveness and efficacy).

A discussion around the conditions used and the potential for participants in both conditions to have used if-then plans is needed. A more passive control condition could have strengthened this study.

What do the authors feel the implications are of conducting this type of research with participants who are not involved or invested in the task? For example, in this sample the participants were not involved in cycling – how do we know that they invested in the task and thus felt that using an if-then plan would be meaningful for them? There is some insightful discussion around this in the limitations, and I wonder whether the researchers could consider how to address this in the analysis?

Reviewer #2: The present study aimed to determine the perceived obstacles and coping strategies during two popular performance tests used to quantify anaerobic and aerobic capacity. Qualitative analysis indicated different main obstacles for endurance performance, e.g. exercise-induced sensations as well as focus on body segments and the cycling technique. In a second session, the same subjects were investigated again and were instructed to use different psychological strategies to improve self-regulation capacity during exercise (goal vs. implementation intention), which, however, did not improve performance significantly.

General comments:

The topic of the present manuscript is of interest, because it investigates ‘psychological’ aspects of endurance performance, which importance has become increasingly clear over the last years. However, from my point of view, there are major issues that should be addressed before this manuscript can be considered for publication in Plos One. These points include the following (see specific comments for details).

(i) In general, the manuscript is too long and uses many words for aspects that should be described more briefly and straight forward. Moreover, plain language is used in some places, which should be avoided.

(ii) The introduction is too long and mixes up general information and detailed study-related information, which in part are better suited in the methods section. Furthermore, the general determining factors of endurance performance should be mentioned to derive the research question. In this context, mechanistic foundations for why exercise-induced obstacles should differ between tasks should be presented.

(iii) Performing both tests in one session introduces the problem of ‘residual fatigue’, which results in a potential bias of the parameters recorded during the second test induced by ‘history-dependent’ modulations.

(iv) The methods lack some information for specific measures, which should be clarified.

(v) The discussion is not appropriate from my point of view and should be amended. The main points are the same as for the introduction.

Specific comments:

1. Introduction:

l45-55: From my point of view, presenting the short everyday example and the aims of the study at this point is not optimal and brings no advantage compared to the “standard” introduction, which tries to summarize the studies related to the topic and present the aims of the study afterwards. Therefore, I recommend to restructure the introduction accordingly.

l60: Please insert a name before reference (2) to optimize the sentence.

l64: Please insert a name before reference (5) to optimize the sentence.

l69: Please insert a name before reference (7) to optimize the sentence. Please check the correctness of sentences, when you change the citations style.

l57-74: From my point of view, the determinates of endurance performance should be introduced at this point, with a focus on ‘perceptual and/or ‘psychological’ aspects. For example, we know, e.g., from the Marcora and Mauger papers, that effort as well as exercise-induced pain perception are key-determinates of endurance performance in humans. It seems obvious, that at least these things should be also self-reported obstacles. After introducing the ‘psychological’ key-determinates of endurance performance, it makes sense to present the data regarding the perceived obstacles of athletes and physically inactive people. Moreover, since endurance performance is, at the end, limited by state fatigue mechanisms, the taxonomy of fatigue provided by Enoka and Duchateau (2016) could also be a theoretical foundation for your experiment. Within this framework perceived fatigability is determined by, e.g. exercise-induced sensations.

l79-81: It is not purposeful for me to present the aims of the study after every subheading. Therefore, please concentrate of the key-elements of your study and present them straight forward in a short introduction, which ends with the aims of the study.

l82-88: You have mixed study characteristics with the general presentation of facts. This is not meaningful to me. At this point you would like to say that the type of obstacle as well as the coping strategy depend on the exercise-intensity, irrespective of your concrete task, i.e. cycling. This is very plausible, since the physiological requirements are different leading, e.g., to higher effort and exercise-induced pain perception during exercise, which can be attributed to (neuro)physiological mechanisms. Therefore, I recommend to state the fact that obstacles as well as strategies might depend on exercise intensity. Thereafter, you should state why this is the case, e.g. metabolic requirements resulting in an increased intensity of different perceptual qualities and a decrease in affective valence (see Venhorst and Ekkekakis papers for these aspects).

l91-100: The same issue as before. You present a lot of information about your study in the introduction. However, this is not meaningful for the readability and understanding of your study. Therefore, I suggest, as mentioned above, that different exercises have, e.g., different metabolic requirements (anaerobic vs. aerobic). Which test and task you have used is not relevant at this point and should be mentioned first in the aims of your study at the end of the discussion and described in detail in the methods.

l96: This is not relevant and in addition not the gold standard to prescribe training intensities.

101-113: This is a general description of the test and its characteristics. You infer that tests would differ regarding the obstacles without a (neuro)physiological description why this should be the case. This is a general limitation of the present manuscript as it is very descriptive and without a mechanistic basis, e.g., for effort and pain perception.

l114-155: This part reads better and adopts a global perspective. However, it is far too long and should be shortened.

l156-165: Here again you present the next aim of the study. As I wrote before, writing a manuscript in this way is not meaningful for me and leads to less understanding. However, I am open for arguments.

l167-189: This part is also too detailed for an introduction. It reads like a ‘study procedure’ section, which should be placed in the methods. Although I prefer to give a short study overview in the introduction, I recommend to shorten this paragraph.

2. Methods

l193: Have you performed a sample size calculation based on the effect sizes of previous studies?

l202: I suggest to present means plus/minus standard deviations, because it is easier to read.

l218: Have you conducted the experiments on the same time of the day?

l229: What do you mean with RPE assessment before and after the test? Effort can only perceived related to a task. Do you mean at the beginning and end of the task? Or retrospectively after the task?

l230: I think ‘elicit’ is not a good term in this regard.

l232-233: Why have you performed the aerobic test on the same day? It is known that perceptual and physiological process are ‘history-dependent’. Therefore, your results for the aerobic activity might be biased due to the prior anaerobic activity. Please comment on this in a limitation of the study section.

l269: I suggest to use ‘time-to-exhaustion’ because the term ‘time to failure’ is critically discussed.

l273: What do you mean with RPE assessment before and after the test? Effort can only perceived related to a task. Do you mean ta the beginning and end of the task? Or retrospectively after the task?

l325: Please insert a name before reference (47) to optimize the sentence. The same applies to l325. Please check and amend this aspect in the whole manuscript. I will not mark it again.

l322: Unfortunately, I am not familiar with the Bayesian t-test. For my understanding, you have a within group comparison for the tasks and afterwards you compare different groups with regard to the coping strategies. Can these dependent and independent measures be mixed up in one statistical model?

3. Results

l349: Please provide a legend for table 1 clarifying the meaning of ‘*’.

l359-361: Here you provide obstacles that might be specific to you recreational active participants. It might be that inactive populations describe other obstacles. These subjects-specific factors should be emphasized in the discussion and limitations of the study section.

l355-391: For some answers you indicate the frequency using percentages but for others not. Please include this information consistently.

l355-469: This part of the results section is very long. Is it perhaps useful to present the statements and frequencies in a table?

Discussion

l523: Since your participants were recreational active, I assume that everybody has cycling experience. I think what you mean is experience with cycling training or something like that. Please specify this aspect.

l523: The word ‘cardiovascular’ is not meaningful in this sentence.

l543: You have to specify your sample. In the introduction you have justified your investigation with the argument that these aspects were often investigated in endurance athletes. Later in the manuscript you have stated that the majority of subjects had no or little experience with cycling. However, here you stated that 16 of them were endurance athletes.

l529-568: As for the introduction, you have to find the way back to a model or the determinates of endurance performance and have to discuss your results in the context of the known factors (e.g. effort and pain perception). This should be done similar to the discussion of boredom. This is a general weakness of your manuscript. You have some parts that are well written and structured, interspaced by paragraphs that are written in plain language and with too many words.

l603: From my point of view, formulations like ‘grain of salt’ are not appropriate for scientific manuscripts.

l617: I think that the heterogeneity of participant’s characteristics is a major limitation. Irrespective of the concrete task (e.g. cycling and running), endurance athletes interpret their physiological signals differentially compared to nonathletes in the different intensity domains. This might have biased/consequences for your outcome.

6. PLOS authors have the option to publish the peer review history of their article (what does this mean?). If published, this will include your full peer review and any attached files.

Reviewer #1: No

Reviewer #2: No

---

## [Author Response · Author response to Decision Letter 0]

29 Jul 2021

#Editor:

Dear Mr. Gruet,

thank you for your helpful review. By incorporating your suggestions, we are confident that the changes we have made to the original version of the paper have improved its quality substantially. We have paid particular attention to streamlining and restructuring the introduction so that it is more straightforward in its argumentation of the aims of the study. As you can also see in detail in the response letter, we have resolved the unclarity regarding the RPE measurements and feel that the manuscript has gained in comprehensibility (e.g., regarding its measures). 

We hope for further positive comments from you, and we would be honored if you were to consider our paper for publication in PLOS ONE.

Thank you for your support!

Sincerely,

Anna Hirsch, Maik Bieleke, Raphael Bertschinger, Julia Schüler und Wanja Wolff

#Reviewer 1:

Thank you for your extensive suggestions and helpful corrections. We have implemented your ideas and argumentations in the introduction and incorporated the research that you suggested (i.e., by Birrer and Morgan 2010, Issue 2). We added necessary information in the method section to ensure greater clarity (Issue 8-10, Issue 12, Issue 16). Thank you for also pointing our what you liked about our manuscript. Especially in the discussion, we included your thoughts and remarks (Issue 5, Issue 6, Issue 11, Issue 13, Issue 14, Issue 17, Issue 19, Issue 20, Issue 21, Issue 22). We feel that the manuscript has benefitted considerably through your revision. For a detailed reponse to your remarks, please see the Response to Reviewers file.

#Reviewer 2:

Thank you for your detailed feedback on our manuscript. We feel that it has substantially benefitted from your suggestions. We hope that you like how we implemented your feedback.

(i) We shortened the manuscript in general and adopted your suggestions, for instance in the results section (Issue 23, S2A-S2C Tables). 

(ii) Also, the introduction has been revisioned and does now describe our research approach more consisely. Method-related information has been repositioned in the method section (Issue 9). We made sure to implement the determining factors of endurance performance (e.g., effort, pain, fatigue; Issue 2) as a foundation for our research question. We gratefully accepted your proposal to present the mechanistic foundations of exercise-related obstacles (Issue 4 and 5).

(iii) Thank you for raising this important limitation. We agree with your evaluation that this influences performance in the second test and addressed this concern specifically in our discussion (Issue 15).

(iv) We apologize for these inclarities and made sure to implement the necessary information (Issue 10, Issue 11, Issue 12, Issue 13, Issue 17 and Issue 19).

(v) We agree with your observation that some aspects could be more thoroughly discussed. This is why we expanded the discussion incorporating your ideas regarding biased parameters in the second test, sample heterogeneity and specification (Issue 15, Issue 21, Issue 24-29).

For a detailed reponse to your remarks, please see the Response to Reviewers file.

---

## [Decision Letter · Decision Letter 1]

22 Sep 2021

PONE-D-21-13120R1Struggles and Strategies in Anaerobic and Aerobic Cycling Tests: A Mixed-method Approach with a Focus on Tailored Self-Regulation StrategiesPLOS ONE

Dear Dr. Anna Hirsch

Thank you for submitting your manuscript to PLOS ONE. After careful consideration, we feel that it has merit but does not fully meet PLOS ONE’s publication criteria as it currently stands. Therefore, we invite you to submit a revised version of the manuscript that addresses the points raised during the review process. Please submit your revised manuscript by 6th October. If you will need more time than this to complete your revisions, please reply to this message or contact the journal office at plosone@plos.org. Please include the following items when submitting your revised manuscript:A rebuttal letter that responds to each point raised by the academic editor and reviewer(s). You should upload this letter as a separate file labeled 'Response to Reviewers'.A marked-up copy of your manuscript that highlights changes made to the original version. You should upload this as a separate file labeled 'Revised Manuscript with Track Changes'.An unmarked version of your revised paper without tracked changes. You should upload this as a separate file labeled 'Manuscript'.If applicable, we recommend that you deposit your laboratory protocols in protocols.io to enhance the reproducibility of your results. Protocols.io assigns your protocol its own identifier (DOI) so that it can be cited independently in the future. For instructions see: https://journals.plos.org/plosone/s/submission-guidelines#loc-laboratory-protocols. Additionally, PLOS ONE offers an option for publishing peer-reviewed Lab Protocol articles, which describe protocols hosted on protocols.io. Read more information on sharing protocols at https://plos.org/protocols?utm_medium=editorial-email&utm_source=authorletters&utm_campaign=protocols.

We look forward to receiving your revised manuscript.

Kind regards,

Mathieu Gruet, Ph.D

Academic Editor

PLOS ONE

Journal Requirements:

Additional Editor Comments (if provided):

Reviewers' comments:

Reviewer's Responses to Questions

**Comments to the Author**

1. If the authors have adequately addressed your comments raised in a previous round of review and you feel that this manuscript is now acceptable for publication, you may indicate that here to bypass the “Comments to the Author” section, enter your conflict of interest statement in the “Confidential to Editor” section, and submit your "Accept" recommendation.

Reviewer #1: (No Response)

Reviewer #2: (No Response)

2. Is the manuscript technically sound, and do the data support the conclusions?

Reviewer #1: Yes

Reviewer #2: Yes

3. Has the statistical analysis been performed appropriately and rigorously? 

Reviewer #1: Yes

Reviewer #2: Yes

4. Have the authors made all data underlying the findings in their manuscript fully available?

Reviewer #1: Yes

Reviewer #2: Yes

5. Is the manuscript presented in an intelligible fashion and written in standard English?

Reviewer #1: Yes

Reviewer #2: Yes

6. Review Comments to the Author

Reviewer #1: Well done to the authors for putting in the effort to addressing the extensive range of comments of both reviewers. The key messages of the study come across a lot more clearly now. I have made a few more comments – the line numbers represent the line numbers of the ‘track changes’ document from page 51 of the full pdf document onwards.

Line 111 – rephrase, ‘to an exercise’ does not read well

Line 123-124 – Is there a supporting reference that can be cited here?

Line 199 – Ensure to provide definition of self-regulation, so that the reader is familiar with the authors’ interpretation of self-regulation

Line 216- Remove ‘the’ before ‘performing’

Line 218-220 – Sentence does not read well

Line 380-386: Can this be rephrased for clarity

Line 521-522: Would suggest to remove interrater reliability and the range of agreement as outlined in my first review. The focus can then shift to the discussion of agreeing to the themes.

Line 560 – Data are plural, change to were

For the thematic analysis – normally the reader would expect to see some direct quotes to illustrate the themes – if there are any particular quotes that can bring a theme come to ‘life’ then I would suggest that the authors bring some (not all) of these back into this section.

Line 692 – Although this is referred to as ambition, arguably this seems to resemble achievement motivation (ego and an approach orientation) rather than ambition?

Line 718 - It is not clear what is meant with anecdotal evidence?

Line 829-831 – I guess what would be relevant to the reader is to know how if-then plans work for less experienced athletes – i.e. what are the applied implications here? Generally a summary of applied implications for the sport psychology practitioner could be stated more clearly.

Line 844 - Would you say that an anaerobic test represents endurance performance? This is not typically considered as such (see also Gastin, 2001)

Line 972-973 – Check sentence structure – grammar.

Reviewer #2: General comments:

The authors did a good job in revising their manuscript. It is much straighter forward now and the readability has improved considerably. Furthermore, it now highlights the “concepts” trying to explain the contributing factors to the psychological determinates of endurance performance as well as the contribution of the authors results to this field. I have some minor suggestions (see below) that should be considered and afterwards the manuscript can be accepted for publication in Plos One.

Specific comments:

1. Introduction:

l50-52: Please provide short examples, why this topic is of scientific and practical importance. You only state that it is, but not why.

l59, 63, …: Please add additional brackets for the references within the brackets.

l105: Please add a comma after test.

l106: I suggest to use “steady” instead of “frequent”.

2. Methods

l170-171: Please use the same description of mean and SD. Here you have both SD in brackets and plus/minus. Please check this aspect throughout the manuscript.

3. Discussion

l587: I suggest to use “steady” instead of “frequent”.

l599: This is not only a limitation from the “exercise physiology perspective” as both physiological and psychological alterations during motor tasks are interdependent and “history-dependent”. Therefore, please adjust the wording accordingly.

7. PLOS authors have the option to publish the peer review history of their article (what does this mean?). If published, this will include your full peer review and any attached files.

Reviewer #1: No

Reviewer #2: No

---

## [Author Response · Author response to Decision Letter 1]

5 Oct 2021

Editor: Issue 1 

-------

Our response: Thank you for your feedback on our manuscript. We made sure to not use retracted papers and especially checked the preprints that we cited (e.g., Wolff et al., 2021, which has been published in the meantime). We assumed that the cited preprints raised the question about retracted papers. If it is another paper that we have not been able to identify so far, please let us know.

According to the reviewers’ feedback, we added the following references to the manuscript:

# 25 Baumeister RF, Vohs KD. Self-regulation, ego depletion, and motivation. Social and Personality Psychology Compass 2007;1(1):115–28. doi: 10.1111/j.1751-9004.2007.00001.x.

#54 Paquette M, Le Blanc O, Lucas SJE, Thibault G, Bailey DM, Brassard P. Effects of submaximal and supramaximal interval training on determinants of endurance performance in endurance athletes. Scand J Med Sci Sports 2017;27(3):318–26. doi: 10.1111/sms.12660. PubMed PMID: 26887354.

#55 Hofman N, Orie J, Hoozemans MJM, Foster C, Koning JJ de. Wingate Test as a Strong Predictor of 1500-m Performance in Elite Speed Skaters. Int J Sports Physiol Perform 2017;12(10):1288–92. doi: 10.1123/ijspp.2016-0427. PubMed PMID: 28253027.

##########################

Reviewer 1: General comments:

Well done to the authors for putting in the effort to addressing the extensive range of comments of both reviewers. The key messages of the study come across a lot more clearly now. I have made a few more comments – the line numbers represent the line numbers of the ‘track changes’ document from page 51 of the full pdf document onwards. 

----

Our response: Thank you very much for the positive reception of our revised manuscript and for the further helpful comments you made on the edited version. We feel that the manuscript benefitted again from your review and we hope that you like how we implemented your feedback.

#########################

Reviewer 2: General comments:

The authors did a good job in revising their manuscript. It is much straighter forward now and the readability has improved considerably. Furthermore, it now highlights the “concepts” trying to explain the contributing factors to the psychological determinates of endurance performance as well as the contribution of the authors results to this field. I have some minor suggestions (see below) that should be considered and afterwards the manuscript can be accepted for publication in Plos One. 

----

Our reponse: Thank you for your positive evaluation of our revision and your further constructive comments below. We thankfully implemented your final feedback on our manuscript.

---

## [Decision Letter · Decision Letter 2]

13 Oct 2021

Struggles and Strategies in Anaerobic and Aerobic Cycling Tests: A Mixed-method Approach with a Focus on Tailored Self-Regulation Strategies

PONE-D-21-13120R2

Dear Dr. Anna Hirsch

We’re pleased to inform you that your manuscript has been judged scientifically suitable for publication and will be formally accepted for publication once it meets all outstanding technical requirements.

Kind regards,

Mathieu Gruet, Ph.D

Academic Editor

PLOS ONE

Additional Editor Comments (optional):

I would like to congratulate you and your co-authors for this very nice study and manuscript.

Best regards

MG

Reviewers' comments:

Reviewer's Responses to Questions

**Comments to the Author**

1. If the authors have adequately addressed your comments raised in a previous round of review and you feel that this manuscript is now acceptable for publication, you may indicate that here to bypass the “Comments to the Author” section, enter your conflict of interest statement in the “Confidential to Editor” section, and submit your "Accept" recommendation.

Reviewer #1: All comments have been addressed

Reviewer #2: All comments have been addressed

2. Is the manuscript technically sound, and do the data support the conclusions?

Reviewer #1: Yes

Reviewer #2: Yes

3. Has the statistical analysis been performed appropriately and rigorously? 

Reviewer #1: Yes

Reviewer #2: Yes

4. Have the authors made all data underlying the findings in their manuscript fully available?

Reviewer #1: Yes

Reviewer #2: Yes

5. Is the manuscript presented in an intelligible fashion and written in standard English?

Reviewer #1: Yes

Reviewer #2: Yes

6. Review Comments to the Author

Reviewer #1: Thank you for addressing all the comments - a very interesting paper that I hope will attract interest from a wide audience.

Reviewer #2: (No Response)

7. PLOS authors have the option to publish the peer review history of their article (what does this mean?). If published, this will include your full peer review and any attached files.

Reviewer #1: No

Reviewer #2: No

---

## [Editor Report · Acceptance letter]

19 Oct 2021

PONE-D-21-13120R2 

Struggles and Strategies in Anaerobic and Aerobic Cycling Tests: A Mixed-Method Approach With a Focus on Tailored Self-Regulation Strategies 

Dear Dr. Hirsch:

I'm pleased to inform you that your manuscript has been deemed suitable for publication in PLOS ONE. Congratulations! Your manuscript is now with our production department. 

Kind regards, 

on behalf of

Dr. Mathieu Gruet 

Academic Editor

PLOS ONE